# Structural insights into the mechanism of rhodopsin phosphodiesterase

Tatsuya Ikuta[1], Wataru Shihoya [1✉], Masahiro Sugiura[2], Kazuho Yoshida[2], Masahito Watari[2], Takaya Tokano[3], Keitaro Yamashita [1], Kota Katayama[2,4], Satoshi P. Tsunoda [2,4], Takayuki Uchihashi [3,5], Hideki Kandori [2,4✉] & Osamu Nureki [1✉]

Rhodopsin phosphodiesterase (Rh-PDE) is an enzyme rhodopsin belonging to a recently discovered class of microbial rhodopsins with light-dependent enzymatic activity. Rh-PDE consists of the N-terminal rhodopsin domain and C-terminal phosphodiesterase (PDE) domain, connected by 76-residue linker, and hydrolyzes both cAMP and cGMP in a light-dependent manner. Thus, Rh-PDE has potential for the optogenetic manipulation of cyclic nucleotide concentrations, as a complementary tool to rhodopsin guanylyl cyclase and photosensitive adenylyl cyclase. Here we present structural and functional analyses of the Rh-PDE derived from *Salpingoeca rosetta*. The crystal structure of the rhodopsin domain at 2.6 Å resolution revealed a new topology of rhodopsins, with 8 TMs including the N-terminal extra TM, TM0. Mutational analyses demonstrated that TM0 plays a crucial role in the enzymatic photoactivity. We further solved the crystal structures of the rhodopsin domain (3.5 Å) and PDE domain (2.1 Å) with their connecting linkers, which showed a rough sketch of the full-length Rh-PDE. Integrating these structures, we proposed a model of full-length Rh-PDE, based on the HS-AFM observations and computational modeling of the linker region. These findings provide insight into the photoactivation mechanisms of other 8-TM enzyme rhodopsins and expand the definition of rhodopsins.

[1] Department of Biological Sciences, Graduate School of Science, The University of Tokyo, Bunkyo, Tokyo 113-0033, Japan. [2] Department of Life Science and Applied Chemistry, Nagoya Institute of Technology, Showa-Ku, Nagoya 466-8555, Japan. [3] Department of Physics, Nagoya University, Nagoya 464-8602, Japan. [4] OptoBioTechnology Research Center, Nagoya Institute of Technology, Showa-Ku, Nagoya 466-8555, Japan. [5] Exploratory Research Center on Life and Living Systems (ExCELLS), National Institutes of Natural Sciences, Okazaki 444-8787, Japan. ✉email: wtrshh9@gmail.com; kandori@nitech.ac.jp; nureki@bs.s.u-tokyo.ac.jp

Microbial rhodopsins are photoreceptive membrane proteins with an all-*trans* retinylidene chromophore[1,2] (all-*trans* retinal). All microbial rhodopsins share a seven transmembrane topology, in which all-*trans* retinal chromophore is bound to a conserved lysine residue via Schiff base in the transmembrane helix (TM) 7. Most microbial rhodopsins function as light-driven ion pumps that actively transport various ions[1] ($H^+$, $Na^+$, $Cl^-$, etc.). Some microbial rhodopsins function as light-gated ion channels (channelrhodopsins[3,4]), which are used as optogenetic tools in neuroscience[5,6]. Enzyme rhodopsins are a group of newly discovered microbial rhodopsins[7], found in eukaryotes such as fungi, green algae, and choanoflagellates. Enzyme rhodopsins comprise the N-terminal rhodopsin domain and C-terminal enzyme domain, connected by a linker, and function as light-activated enzymes. Two types of enzyme rhodopsins were initially discovered, the histidine kinase rhodopsins[8] (HKRs) and rhodopsin guanylyl cyclase[9] (Rh-GC). While HKRs function as ATP-dependent light-inhibited guanylyl cyclase[10], Rh-GC from *Blastocladiella emersonii* (*Be*GC1) induces a rapid light-triggered cGMP increase in heterologous cells (~5000-fold) and could be utilized as an optogenetic tool to rapidly manipulate cGMP levels in cells and animals[11,12].

In 2017, rhodopsin phosphodiesterase[13] (Rh-PDE), a type of enzyme rhodopsin, was discovered. Rh-PDE derived from *Salpingoeca rosetta* (*Sr*Rh-PDE) was first identified and exhibits light-dependent hydrolytic activity for both of cAMP and cGMP, whereas it lacks pump or channel activity. Rh-PDE is constitutively active in the dark, and the increase in its light-dependent hydrolytic activity is up to twofold[14]. Moreover, four Rh-PDE homologs were recently discovered in *Choanoeca flexa*[15]. These organisms belong to the choanoflagellates, a group of free-living unicellular and colonial flagellate eukaryotes, considered as the closest living relatives of animals. In response to sudden darkness, Rh-PDEs trigger the coordinated, polarized contraction of *C. flexa* cells, which results in colony inversion. Since the light-dependent phosphodiesterase activity of Rh-PDE is complementary to that of photosensitive adenylyl cyclase[16,17] (PAC) and Rh-GC, *Sr*Rh-PDE has potential for the optogenetic manipulation of cyclic nucleotide concentrations and cell signaling.

*Sr*Rh-PDE consists of the long N-terminal region, transmembrane domain (TMD), including the rhodopsin domain, C-terminal PDE domain, and 76-residue linker region connecting the TMD and PDE domain. The rhodopsin domain of *Sr*Rh-PDE shares relatively low sequence identity with other microbial rhodopsins (about 20%)[18]. Moreover, as in the other enzyme rhodopsins, *Sr*Rh-PDE is predicted to have eight transmembrane helices, in which the extra helix TM0 is at the N-terminus of the rhodopsin domain[19]. To date, the structures of the TMDs and the full-length enzyme rhodopsins have not been determined. Thus, the existence and functional role of TM0 have remained unclear. While the structure of the PDE domain of *Sr*Rh-PDE was determined[19], little is known about the overall domain architecture and photoactivation mechanism of the full-length Rh-PDE. Here, we present crystal structures of *Sr*Rh-PDE–TMD with and without the linker region, at 3.5 and 2.6 Å resolutions, respectively. The structures and mutational analyses revealed a 8-TM topology and the functional mechanism of the extra TM. We also determined the crystal structure of the PDE domain with the linker region, at 2.1 Å resolution. Based on these three structures, we propose a model of the full-length Rh-PDE structure and the mechanism of the light-dependent enzyme activation, with the support of HS-AFM observations and computational modeling of the linker region.

## Results

### Overall structure of Rh-PDE–TMD. 
Initial crystallization trials of the full-length Rh-PDE were unsuccessful. Thus, we expressed, purified and crystallized the TMD of Rh-PDE (Rh-PDE–TMD, residues 33–316). Crystals were obtained with the lipidic cubic phase (LCP) method (Supplementary Fig. 1a, b), and the diffraction data sets were collected with an automated data collection system, ZOO[20]. In total, 271 datasets were merged with the KAMO software[21]. The structure was determined at 2.6 Å resolution by molecular replacement, using the structure of bacteriorhodopsin (BR, PDB ID: 1M0M[22]) (Supplementary Table 1). The crystallographic asymmetric unit contained two molecules (mol A and mol B). The overall architectures of these two molecules are essentially identical (root mean square deviation of 0.22 Å over all Cα atoms, Supplementary Fig. 1c), and thus we focused on the mol A structure, in which residues 47–310 are observed.

The Rh-PDE–TMD structure consists of eight transmembrane helices (TM0–7). TM1–7 adopt a canonical rhodopsin-like topology with covalently linked retinal at Lys296 in TM7, as in the other microbial rhodopsins. Moreover, one extra transmembrane helix (TM0) is observed at the N-terminus, as previously predicted[19] (Fig. 1a, b). TM0 is connected to TM1 via an extracellular loop (ECL0), folding into a lateral short helix. Owing to the 8-TM topology, both the N and C termini face the intracellular side, consistent with the previous immunofluorescence staining analysis[19].

A previous crystallographic study[19] showed that the PDE domain in Rh-PDE itself forms a dimer, as in other PDEs[23–25], suggesting that the full-length Rh-PDE also functions as a dimer. In the Rh-PDE–TMD structure, the two protomers in the asymmetric unit form a dimer. The interfacial interactions occur in the intracellular halves of TM1 and TM7 of each molecule, whereas two monoolein molecules fill the dimer interface at the extracellular half (Fig. 1c). Thus, Rh-PDE–TMD forms a loosely packed dimer, as compared with channelrhodopsin-2 (ChR2) and heliorhodopsin (HeR) (Supplementary Fig. 2). The TM7 kink at Lys296 is larger than that of BR, resulting in a rigid dimer interface with a hydrogen bond between Asn305, Glu309, and Asn113 (Fig. 1d). Size-exclusion chromatography with multi-angle laser light scattering (SEC–MALLS) analysis showed that Rh-PDE TMD itself forms a dimer (Supplementary Fig. 1d, e), suggesting that the dimerization at TM1 and TM7 reflects a physiological state. The residues in the dimer interface are conserved in the recently discovered Rh-PDE homologs[15] (Supplementary Figs. 3 and 4a), suggesting that dimer formation at TM1 and TM7 is a conserved structural feature in the Rh-PDE homologs, which has not been observed in the other microbial rhodopsins.

### Detailed structural comparisons with BR and ChR2. 
To examine the structure-function relationship of Rh-PDE, we compared the rhodopsin domain (TM1–7) of Rh-PDE with bacteriorhodopsin[26] (BR) and ChR2[27], which are representative microbial rhodopsins. Rh-PDE has 24.4% and 19.3% sequence identities with BR and ChR2, and the Rh-PDE structure superimposes well on those of BR and ChR2 (RMSD values of 1.74 and 2.16 Å over all Cα atoms, respectively) (Fig. 2a, b). The ECL1 between TM2 and TM3 forms two anti-parallel β strands, as in BR and ChR2. The TM helices of Rh-PDE are aligned with those of BR, while the orientation of TM1 is quite different between Rh-PDE and ChR2. Moreover, TM7 of ChR2 is extended by two α-helical turns, as compared with that of Rh-PDE. Taken together, the rhodopsin domain of Rh-PDE more closely resembles that of BR, rather than ChR2.

Lys296 at TM7 forms a covalent bond with a retinal-Schiff base (RSB) of all-*trans* retinal (ATR), as in the other microbial rhodopsins (Fig. 2c–e). Asp85 and Asp212 in BR connect to the

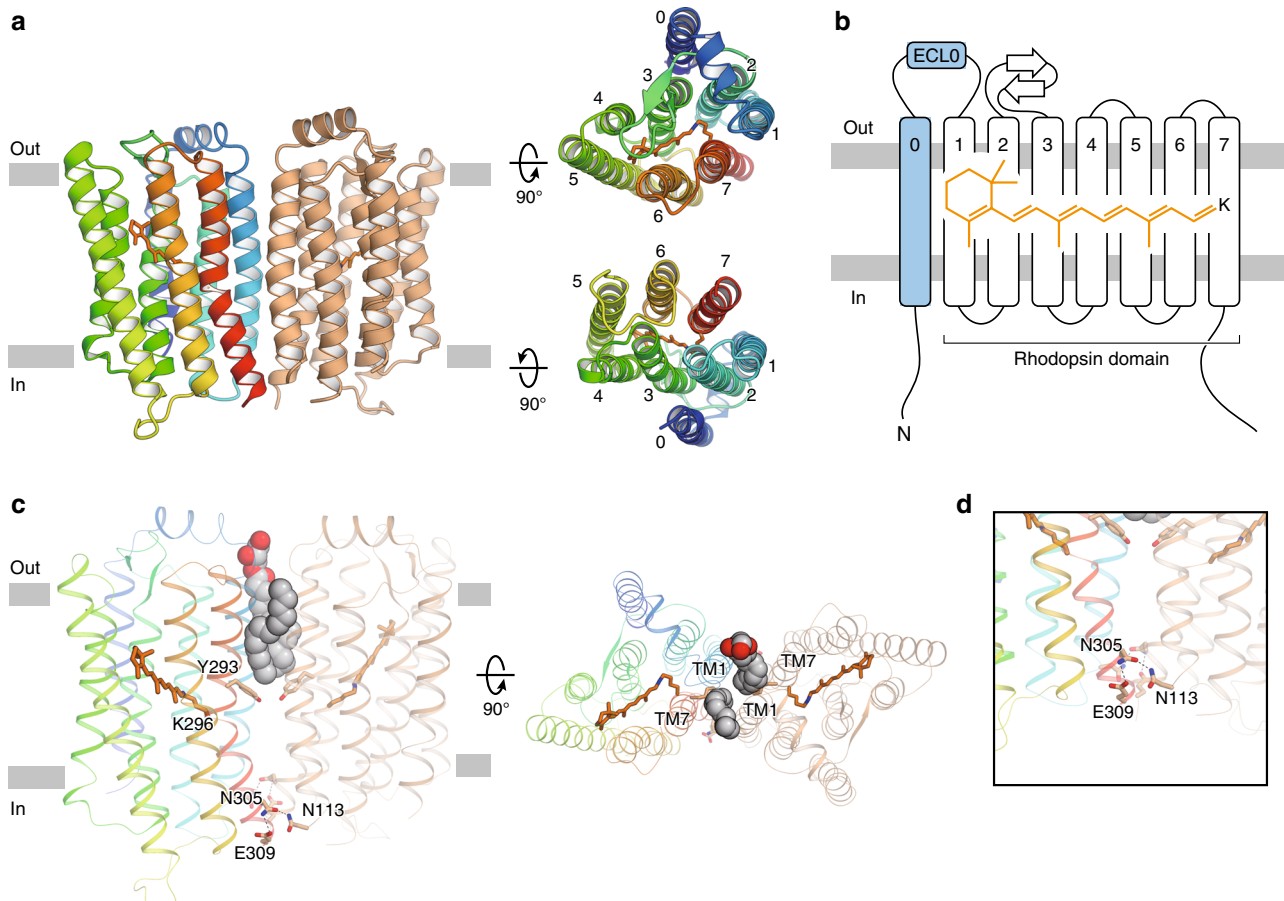

**Fig. 1 Overall structure. a** Ribbon diagrams viewed from the membrane plane (left), the extracellular side (upper right) and the intracellular side (lower right). One protomer is colored rainbow and the other is colored orange. The retinal chromophores are shown as orange stick models. **b** Schematic representation of the ribbon diagram. **c** Monoolein molecules and residues at the dimer interface, viewed from the membrane plane (left) and the extracellular side (right). Monoolein molecules are shown as spheres with gray carbon atoms and red oxygen atoms. **d** Close-up view of the residues on the intracellular side.

RSB via a water molecule and work as counterions (Fig. 2c), which is one of the most prominent factors in color tuning[1]. Rh-PDE conserves these acidic residues as Glu164 and Asp292. Previous FTIR and E164Q mutant studies proposed that Asp292 is the unique counterion, and Glu164 does not function as a counterion because it is always protonated[18]. These studies also suggested that there is no water molecule around the Asp292 counterion. In the Rh-PDE structure, Asp292 directly hydrogen bonds with the RSB, and the protonated Glu164 directly hydrogen bonds with Asp292 (Fig. 2d). Moreover, there is no water molecule around the counterion in the current structure. Overall, the interaction between the RSB and counterion is in excellent agreement with the previous FTIR study.

The retinal polyene chain interacts with the hydrophobic residues in the rhodopsin domain of Rh-PDE, as in BR and ChR2 (Fig. 2f–h). Notably, the bond between C11 and C12 is surrounded by aromatic residues in Rh-PDE (Trp165/Phe216/Trp264/Phe267) (Fig. 2g). These aromatic residues are highly conserved in the other Rh-PDE homologs, except for Phe267 (Supplementary Fig. 3). The phenylalanine is replaced with a histidine in *C. flexa* Rh-PDE 3, and it is surrounded by aromatic residues. These residues play a critical role in the exclusive 13-*cis* isomerization of the retinal chromophore via steric hindrance. Notably, in the vicinity of the β-ionone ring, Trp189 in BR or Phe230 in ChR2 is replaced with Glu271 in Rh-PDE, which is conserved over all Rh-PDE homologs. The hydrophilic glutamic

acid flips and does not participate in the β-ionone ring recognition, thus weakening the hydrophobic interactions around it.

The absorption maximum of Rh-PDE is 492 nm[13], which is an intermediate value between those of BR (568 nm[28]) and ChR2 (480 nm[3]). The rhodopsin absorption maximum depends on the energy gap between the ground and excited states; that is, the interaction between the retinal and protein[1,29]. The absorption maximum will be red-shifted when the energy gap is small, while blue-shifted when the gap is large. The energy at the ground state greatly depends on the counterions around the RSB. Asp292 in Rh-PDE (corresponding to Asp212 in BR) directly hydrogen bonds with the RSB (Fig. 2d) and the energy at the ground state is lower than that of BR, as in ChR2 (Fig. 2e), explaining blue shift of Rh-PDE and ChR2 as compared to BR. The energy at the excited state is affected by the environment around the retinal polyene chain. In Rh-PDE, the hydrophilic Glu271 around the β-ionone ring reduces the energy level of the excited state as compared to BR and ChR2, explaining red shift of Rh-PDE as compared to ChR2. Because the interaction between the retinal and counterion is the most essential factor for the absorption, the largest energy gap will be ChR2, followed by Rh-PDE and BR (Fig. 2i). This is consistent with the present finding that the absorption maximum of Rh-PDE is more blue-shifted than BR and more red-shifted than ChR2.

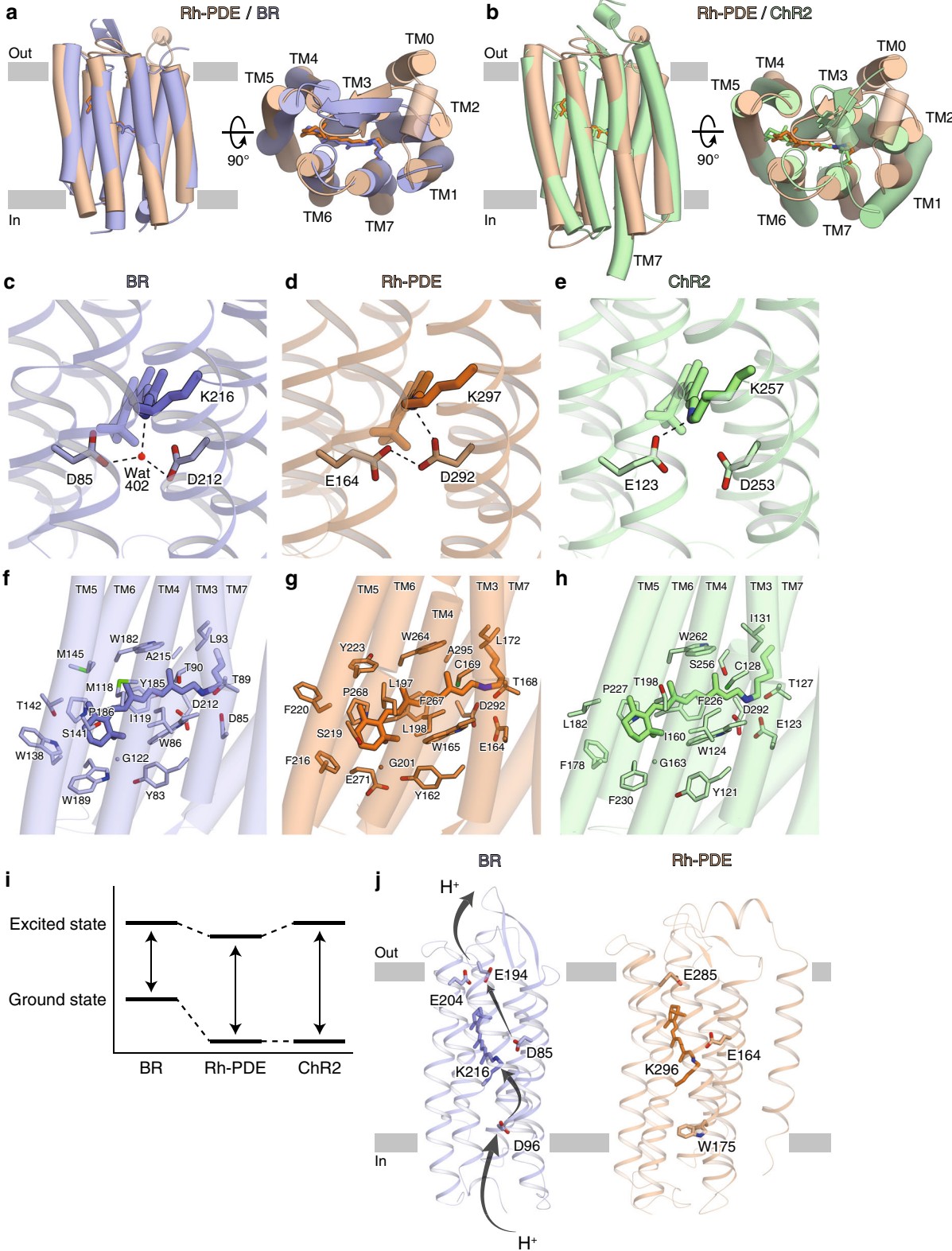

**Fig. 2 Structural comparisons with BR and ChR. a**, **b** Rh-PDE structure superimpositions on the BR structure (**a** PDB ID: 1C3W) and the ChR2 structure (**b** PDB ID: 6EID). Rh-PDE, BR, and ChR2 are colored orange, blue and green, respectively. **c–e** Residues around the RSBs of BR (**c**), Rh-PDE (**d**), and ChR2 (**e**). Dashed lines represent hydrogen bonds. **f–h** Structures of the retinal binding pockets of BR (**f**), Rh-PDE (**g**), and ChR2 (**h**). **i** Schematic diagram of the ground and excited states. **j** Residues contributing to proton transfer in BR (left) and corresponding residues in Rh-PDE (right).

Unlike other typical microbial rhodopsins, Rh-PDE lacks pump or channel activity[18]. All the microbial rhodopsins with pump or channel activity permeate proton, coupled with their respective ion conductance. In BR, protons are passed by the acidic residues and RSB[1,30] (Asp96, Lys216, Asp85, and Glu204 in Fig. 2j). However, Asp96 in BR is replaced with Trp175 in Rh-PDE. Trp175 closes a putative ion conduction pathway, preventing the transport of protons from the cytosol. Moreover, Glu164 is always protonated and forms a hydrogen bond with Asp292; thereby, Glu164 cannot donate or accept any protons. These two differences may preclude the pump and channel activities of Rh-PDE.

**Functional role of TM0.** At the N terminus of the rhodopsin domain, residues 50–73 form an additional transmembrane α-helix, TM0, consistent with the previous prediction[19] (residues 52–75). TM0 exists on the exterior of the rhodopsin domain, and is located between TM2 and TM3 (Fig. 3a). Trp63 in TM0 hydrogen bonds with Ser136 in TM2, and the other residues form extensive hydrophobic interactions with TM2 and TM3. The short helix ECL0 connects TM0 with TM1 and lies over ECL1. In ECL0, the bulky residues protrude toward the extracellular side (Glu77/Arg80/Arg81/Lys85), whereas Ala78 is oriented toward ECL1, preventing a steric clash. TM0 and ECL0 are unique structural features of the enzyme rhodopsin such as Rh-PDE. Our structure is the first to show the 8-TM rhodopsin.

Owing to TM0, the N-terminal region (residues 1–46) faces the cytosol, and is predicted to have no secondary structure[19]. To examine the functional role of the N-terminal residues, we truncated the N-terminal 32 residues (Δ32), as in the crystallization construct (Fig. 3a). We first measured the rhodopsin absorption of the DDM-solubilized Rh-PDE by a hydroxylamine bleach assay (Fig. 3b). The Δ32 mutant showed a rhodopsin-related absorption maximum at 491 nm, similar to that of the wild type (492 nm). However, the peak height was only a quarter of that of the wild type. As the spectrum suggests, a western blot of the RHO 1D4 tag-fused Δ32 mutant showed only a quarter of the wild type in the expression level (Fig. 3c, d). Therefore, the N-terminal region plays an important role in the proper expression of Rh-PDE. We next measured the enzymatic photoactivity by an HPLC analysis of the products (Fig. 3e, f). The wild-type Rh-PDE showed three and twofold light-induced productions of cAMP and cGMP, respectively, while the Δ32 mutant showed twofold production, indicating that the cytosolic N-terminal residues facilitate the tight light regulation of the cytosolic PDE activity.

To examine the functional role of TM0, we next truncated TM0 (Δ76) and ECL0 (Δ88) and performed the functional analyses. Both mutants showed no rhodopsin-related absorption in the DDM micelles (Fig. 3b), and their dark activities increased significantly (Fig. 3e, Supplementary Fig. 4b). Nevertheless, these mutants showed certain expression levels (Fig. 3c, d) and their enzyme activities retained slight light dependency (Fig. 3e, f), suggesting that the retinal can bind to the truncated mutants. These results indicated that the TM0 truncation destabilizes the retinal chromophore, which dissociates when the mutants are solubilized in the DDM micelles. Since TM0 interacts with TM2 and TM3 (Fig. 3a), it stabilizes the overall folding of the TMD, indirectly facilitating the interactions with the retinal chromophore, and thus playing a crucial role in the enzymatic photoactivity. Taken together, TM0 is necessary for the structure and functions of Rh-PDE, unlike the other microbial rhodopsins.

**Structural model of the full-length Rh-PDE.** The above Rh-PDE–TMD structure revealed the architecture of the enzyme rhodopsin, but the TMD and PDE domain are connected by a linker region (residues 311–386). To obtain structural insight into the full-length Rh-PDE, we determined the 3.5 Å resolution crystal structure of Rh-PDE–TMD with the linker region (TMD–Linker, residues 33–378) (Fig. 4a, Supplementary Fig. 5a, b, and Supplementary Table 1). The TMD–Linker structure superimposes well on the TMD structure (RMSD value of 0.84 Å over all Cα atoms) (Supplementary Fig. 5c). While the crystal packing is different between the TMD and TMD–Linker structures, both form a similar dimer interface (Supplementary Fig. 5d, e). In one protomer, the C-terminal residues are observed up to Trp353 (Fig. 4a and Supplementary Fig. 5f). Residues 311 to 332 adopts a straight α-helix continuous from TM7, while residues 333–353 adopt a short loop and α-helix, turning back to the TMD. In the other protomer, the C-terminal residues are observed up to His320 and also adopt a straight α-helix. In the dimer, the α-helices in the linker regions cross each other, suggesting that the PDE dimer is swapped between the TMD dimer. Although the crystal packing might affect the conformation of the linker region (Supplementary Fig. 5h), we can interpret its structure and connectivity with the TMD.

We next determined the 2.1 Å resolution crystal structure of the PDE domain with the linker region (Linker–PDE, residues 347–703) (Fig. 4b and Supplementary Table 1). The Linker–PDE structure superimposes well on the structures of the PDE domains of Rh-PDE[19] and PDE9[24] (RMSD value of 0.43 and 1.26 Å over all Cα atoms, respectively), which have similar dimer interfaces (Supplementary Fig. 6a, b). The N-terminal residues Lys380–Met386 are newly observed as compared with the reported structure (Fig. 4b and Supplementary Fig. 6a), while residues 347–379 are disordered. The Ala385 backbone amide forms a hydrogen bond with the Glu478 sidechain, fixing the N-terminal residues (Fig. 4c) and extending toward the dimer interface.

The TMD–Linker and Linker–PDE structures provided insights into their connectivity, but there is no observed density to assign residues 354–379 in the linker region (Fig. 4d). Therefore, we created an ab initio model of the linker region (residues 311–386) using the ab initio modeling program, Rosetta[31,32] (Fig. 4e). The model comprises three straight α-helical bundles, in which the N- and C-termini face in opposite directions. The first helix in the model (residues 313–333) is similar to the α-helix extending from TM7 (residues 309–332). Given that the linker region connects the TMD and cytosolic PDE domain, the model is valid as a connecting linker. For further validation of the model, we performed accelerated molecular dynamics (aMD) simulations[33,34] for 150 ns, starting from the Rosetta model, which explore a wider structural space than conventional MD (cMD) simulations. The aMD simulations were performed with boosting of the dihedral potential (aMDd), total potential (aMDT), and both (aMDdual) to test the stability of the model. The final structures of the aMDd and aMDT simulations showed stable α-helices with RMSD values of 3.65 and 3.12 Å over all Cα atoms, respectively (Supplementary Fig. 7a, b). The final structure of the aMDdual simulation, with the highest boost energy, presented a relatively high RMSD value of 6.01 Å over all Cα atoms. Nevertheless, the first helix and some parts of the second and third helices were stable (Fig. 4f and Supplementary Fig. 7c). The C-terminal residues 380–386 in the third α-helix were unwound, as similarly in our Linker–PDE structure. Overall, the results of the aMD simulations are consistent with our crystal structures, suggesting that the linker region forms α-helical bundle.

To obtain structural insight into the full-length Rh-PDE, we reconstituted the purified full-length Rh-PDE in a lipid bilayer and performed high-speed atomic force microscopy (HS-AFM) observations. We detected two major structures, containing large

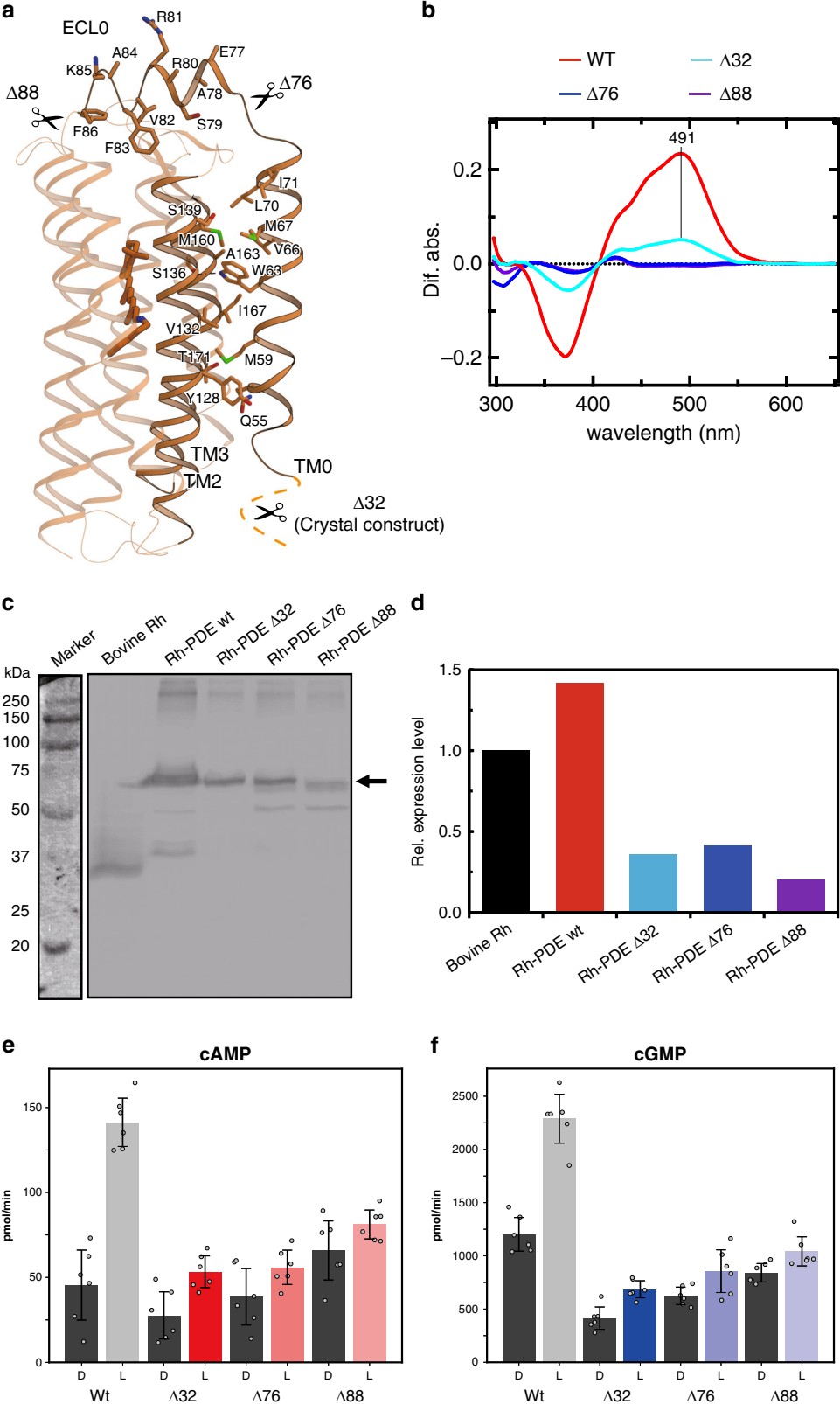

**Fig. 3 Structure and function of TM0. a** Residues at the interface between TM0 and the rhodopsin domain. **b** Absorption spectrum with hydroxylamine bleach of the N-terminal truncated mutants. **c** Representative immunoblots indicating expression levels of bovine Rh (~35 kDa), Rh-PDE wt (~75 kDa), Rh-PDE Δ32 (~72 kDa), Rh-PDE Δ76 (~67 kDa), and Rh-PDE Δ88 (~65 kDa) in detergent solubilized fraction isolated from HEK293T cells and detected with the anti-Rho C-terminal 1D4-tag antibody. Source data are provided as a Source Data file. **d** Band intensities in (**c**) were quantified and calculated the relative expression level against to bovine Rh obtained with imageJ software ($n = 1$). **e, f** Photoactivities of truncated mutants with cAMP (**e**) and cGMP (**f**). Data are presented as mean values ± standard deviation (SD) ($n = 6$ independent experiments).

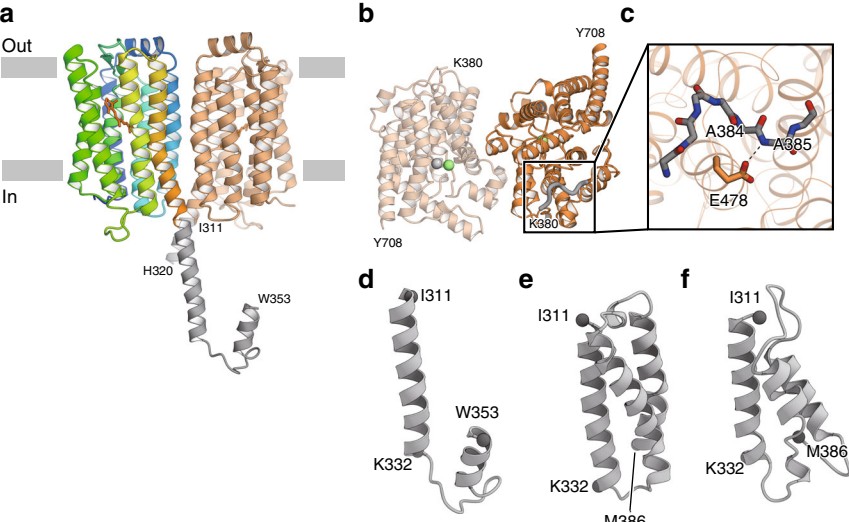

**Fig. 4 Structures with the linker region. a** Overall structure of Rh-PDE TMD–Linker. The linker region is colored gray. **b** Overall structure of Rh-PDE Linker–PDE. The newly observed region, as compared to the previous structure (PDB ID: 5VYD), is shown as gray bold coils. Zinc and magnesium ions are colored gray and green, respectively. **c** Glu478 and the backbone of the newly observed region are shown as stick models. Dashed lines show hydrogen bonds. **d** Ribbon diagrams of the linker structure of Rh-PDE TMD–Linker. **e** Ribbon diagrams of the linker region structure, modeled with Rosetta. **f** Ribbon diagrams of the final structure from the aMDdual simulation.

and small dimer rods (Fig. 5a and Supplemental Videos 1–3). Since the TMD is wrapped in the lipid molecules, it could be observed as a larger rod. The small rods are in close agreement with the simulated AFM images constructed from the crystal structure (Supplementary Fig. 8a). These observations indicated that the large and small rods correspond to the TMD and PDE domain, respectively. Both the TMD and PDE domain are observed as dimers, consistent with the crystal structures. Between the TMD and PDE domain, weak density was observed, corresponding to the linker region. The TMD and PDE domain do not form any direct interactions.

Finally, we combined the three structurers of the TMD–Linker, Linker–PDE, and aMDdual into the full-length model using the program, PyMOL (Fig. 5b). The aMDdual structure was rotated and translated to align its first helix with the α-helical linker in the TMD–Linker structure. The distance between centers of the TMD and PDE domain was adjusted to that observed in the HS-AFM images (Supplementary Fig. 8b–d). In this model, TM7 is connected to the PDE domain via the linker regions crossing each other, resulting in the swapped position between the TMD and PDE domain. The linker region adopts a dimeric, four-helical coiled-coil. This structure resembles a HAMP domain[35], which is found in bacterial membrane sensors and transmit conformational changes in the periplasmic ligand-binding domain to cytoplasmic signaling modules. Although the linker region does not contain the HAMP motif, it may dimerize and transmits the conformational change of the TMD to the PDE domain, as the HAMP domain.

**Insight into photoactivation.** In our model of the full-length Rh-PDE, there is no direct contact between the TMD and PDE domain. Thus, the retinal isomerization may displace TM7, followed by the movement of the linker region, thereby inducing the conformational change in the PDE domain for the photoactivation. There are many bulky, hydrophobic residues between the intracellular side of TM7 and the retinal chromophore (Fig. 5c). Mutations of these residues markedly reduced the enzymatic photoactivity in a GloSensor assay[13] (Fig. 5d), which measures luminescence coupled with cAMP binding to the engineered luciferase. Notably, the mutations of Leu299 and Met303, which

are oriented toward the transmembrane helices, reduced the photoactivity. By contrast, the mutation of Ile302, which faces the membrane environment, did not affect the photoactivity. These results suggest that the retinal isomerization physiologically induces the structural changes of TM7 for the photoactivation. The mutations of Met301 and Val304, which extend toward the dimer interface, also reduced the photoactivity. SEC–MALLS analysis showed that these mutations reduce the ratio of dimers of Rh-PDE TMD (Supplementary Fig. 1f), suggesting that the dimeric interactions between TM1 and TM7 facilitate the light-induced structural change of TM7.

Moreover, the intracellular loops contact the intracellular end of TM7. In BR, ICL3 is short and does not form any contacts with other TMs (Fig. 5e). In Rh-PDE, ICL3 contains the PPPPLP sequence and forms a Pro-rich right-handed helical loop (Fig. 5f, g). ICL3 and ICL1 interact with the intracellular end of TM7 to limit the movement. To investigate the functional role of ICL3, we replaced PPPPLP with the flexible linker GGGGSG. This mutant showed about 1.5-fold light-induced productions of cAMP and cGMP (Fig. 5h), which are lower than those of the wild type, indicating that the Pro-rich loop in ICL3 is also associated with the enzymatic photoactivity. Overall, the intracellular half of TM7 forms extensive intramolecular interactions to allow its light-induced structural change. Nevertheless, in the HS-AFM movies, the PDE dimer and linker region were flexible and not tightly fixed to the TMD (Supplemental Video 3), suggesting that the PDE domain is loosely coupled to the TMD. This flexible nature reflects the high dark activity of Rh-PDE.

## Discussion

Our Rh-PDE–TMD structure revealed that the Rh-PDE comprises eight transmembrane helices. TM1–7 adopt the canonical topology of the rhodopsins with the retinal chromophore, and TM0 is located on the outside of TM2 and TM3. Structural comparisons with BR and ChR2 revealed the basic property as a rhodopsin. The structure-guided functional analyses elucidated that TM0 and the cytosolic N-termini facilitate the light dependency of the enzyme activity. Moreover, the TMD–Linker and Linker–PDE structures provided structural insight into the connectivity of the TMD, linker region, and PDE domain. Based on

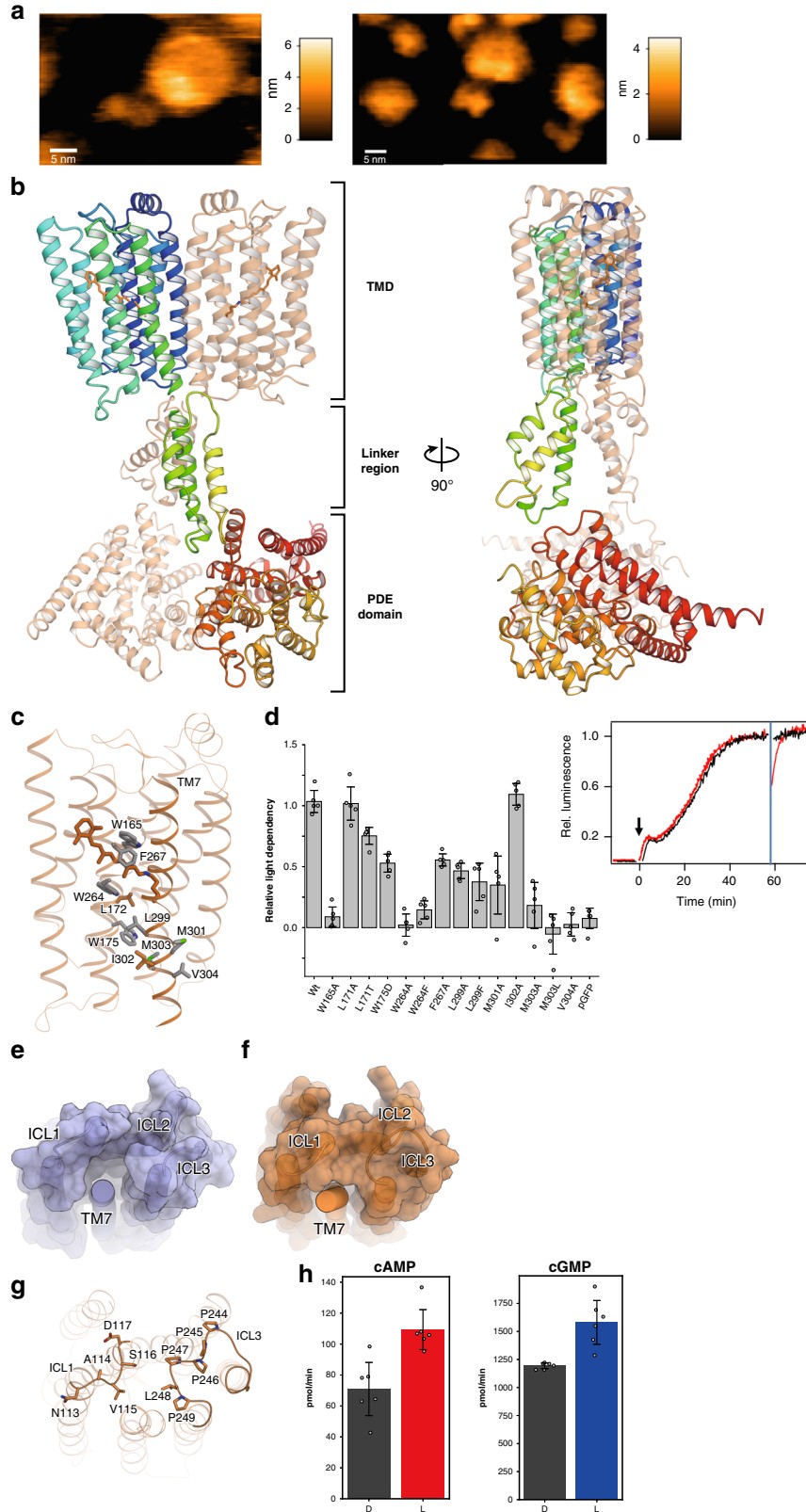

**Fig. 5 Photoactivation mechanism of Rh-PDE. a** HS-AFM images of the full-length Rh-PDE. **b** Ribbon diagrams of the full-length Rh-PDE model. One protomer is colored rainbow and the other is colored light orange. **c** Bulky residues between the intracellular side of TM7 and the retinal chromophore. **d** Relative activities of mutants, measured with a GloSensor assay. Data are presented as mean values ± standard deviation (SD) ($n = 5$ for I302A; $n = 6$ for the other, independent experiments). **e**, **f** Molecular surfaces of BR (**e**) and Rh-PDE (**f**), drawn without TM7. **g** Residues of ICL1 and ICL3 in Rh-PDE. **h** Photoactivities of the ICL3 mutant with cAMP (left) and cGMP (right). Data are presented as mean values ± standard deviation (SD) ($n = 6$ independent experiments).

these structures, we proposed the structural model of the full-length Rh-PDE, which was validated by the HS-AFM analysis.

Rh-PDE forms a dimer in the crystal and supposed to form a dimer under the physiological condition, while Rh-GC catalytic domain is a monomer in the crystal and guessed to be in a monomer/dimer equilibrium under the physiological condition[36]. Rh-GC shows ~5000-fold light dependency with weak dark activity, which is much higher than that of Rh-PDE (~2-fold). The high monomer ratio in the equilibrium may be associated with the weak dark activity of Rh-GC.

Rhodopsins commonly have seven TMs and the retinal chromophore. Recently, a distinct and abundant group of microbial rhodopsins, HeRs, has been discovered[37–39]. Although HeRs share low sequence identity with the other rhodopsins and adopt the inverted membrane topology, the crystal structures revealed that HeR comprises seven TMs with the retinal chromophore, as in the other rhodopsins[40–42]. Unlike these rhodopsins, the enzyme rhodopsins have long N-termini, and are predicted to have 1–2 additional transmembrane helices[11,12,43], while the existence of TM0 and its function still have remained to be elucidated. The current Rh-PDE structures revealed that TM0 certainly exists and plays key roles in the stable retinal binding to TM1–7 and photoactivation (Fig. 3a, b). The sequence alignment of the representative enzyme rhodopsins indicated the good alignment of their rhodopsin domains (Supplementary Fig. 9), whereas their N-terminal residues share low sequence identity. In BeRh-GC, the cytosolic N-termini facilitate the light dependency of the enzyme activity, as in Rh-PDE[14], suggesting that the functions of TM0 and the cytosolic N-termini are conserved in the enzyme rhodopsins. Therefore, the evolution to 8-TM topology may have been necessary for the function as a light-dependent enzyme rhodopsin. The existence of the 8-TM rhodopsin implies that diverse topologies of rhodopsins exist in nature.

## Methods

**Preparation of TMD and TMD–Linker of Rh-PDE.** The Rh-PDE (full) gene (NCBI Gene ID: 16078606) was synthesized after human codon optimization (GenScript)[13]. The Rh-PDE (TMD) gene (encoding aa 33–316) and Rh-PDE (TMD–Linker) gene (encoding aa 33–378) were subcloned into the pEG BacMam vector, with a C-terminal GFP-His$_8$ tag and a tobacco etch virus (TEV) cleavage site, and both genes were expressed in HEK293S GnTI$^-$ cells[44]. Cells were collected by centrifugation (5,000×g, 10 min, 4 °C) and disrupted by sonication in buffer A (20 mM Tris, pH 8.0, 200 mM NaCl, 20% glycerol) supplemented with 100 μM all-trans retinal (Sigma). The membrane fraction was collected by ultracentrifugation (185,500×g, 1 h, 4 °C), and solubilized for 1 h at 4 °C in buffer B (20 mM Tris, pH 8.0, 200 mM NaCl, 10 mM MgCl$_2$, 20% glycerol, 1% n-dodecyl-β-ᴅ-maltoside, DDM (Calbiochem), 0.2% cholesteryl hemisuccinate, CHS (Sigma)). Insoluble materials were removed by ultracentrifugation (185,500×g, 20 min, 4 °C). The supernatant was incubated with TALON Metal Affinity Resin (Clontech) for 30 min at 4 °C. The resin was washed with buffer C (20 mM Tris, pH 8.0, 500 mM NaCl, 10 mM MgCl$_2$, 15 mM imidazole, 10% glycerol, 0.1% 2,2-didecylpropane-1,3-bis-β-ᴅ-maltopyranoside, LMNG (Anatrace), 0.01% CHS), and eluted with buffer C containing 200 mM imidazole. The eluate was treated with TEV protease and dialyzed against buffer D (20 mM Tris–HCl, pH 8.0, 500 mM NaCl, 10 mM MgCl$_2$, 10% glycerol). The protease and cleaved GFP-His$_8$ tag were removed with TALON resin, and the protein was further purified by SEC on a HiLoad 16/600 Superdex 200 pg column (GE Healthcare), equilibrated with buffer E (20 mM Tris, pH 8.0, 150 mM NaCl, 10% glycerol, 0.01% LMNG, 0.001% CHS). The peak fractions were collected and frozen until the crystallization.

**Crystallization of TMD and TMD–Linker of Rh-PDE.** The purified protein was concentrated to 10 mg /mL and mixed with monoolein (Nu-Chek Prep) in a 2:3 protein to lipid ratio (w/w)[45]. Aliquots of the protein-LCP mixture were dispensed onto 96-well glass plates and overlaid with the precipitant solution, using a Gryphon LCP (Art Robbins Instruments, LLC). Crystals of Rh-PDE (TMD) were obtained in 29–35% (w/v) PEG400, 100 mM Na-citrate, pH 5.0, 100 mM KSCN and 10 mM ZnCl$_2$, and crystals of Rh-PDE (TMD–Linker) were obtained in 25% (w/v) PEG500DME, 100 mM Na-citrate, pH 5.0, 100 mM NaK-tartrate. All crystals were incubated at 20 °C for 1 week in the dark. The crystals were harvested using micromounts (MiTeGen) and were flash-cooled in liquid nitrogen without any additional cryoprotectants.

**Structure determination of the TMD and TMD–Linker of Rh-PDE.** X-ray diffraction datasets were collected on beamline BL32XU at SPring-8, with 10× v15 (for TMD) or 5 × 5 (for TMD–Linker) μm$^2$ (width × height) micro-focused beams and an EIGER X 9 M detector (Dectris). Small wedge data sets (10° per crystal for TMD or 5° for TMD–Linker) were collected manually or automatically with the ZOO system[20], an automatic data collection system developed at SPring-8. The loop-harvested microcrystals were identified by raster scanning and subsequently analyzed by SHIKA[46]. The collected images were processed with KAMO[21] with XDS[47]. 864 and 74 datasets of the TMD and TMD–Linker were indexed with the consistent unit cell parameters. For TMD, 271 datasets were finally merged after hierarchical clustering based on correlation coefficients of intensities and outlier rejections implemented in series in kamo.multi_merge and XSCALE. For TMD–Linker, 58 datasets were merged after outlier rejections. The structure of TMD was determined by molecular replacement using Phaser[48]. The search model was prepared using HHpred[49] and phenix.mr_model_preparation[50] using the structure of BR (PDB ID: 1M0M) with a sequence identity of 22%. The resultant structure (TFZ score of 3.3) was further improved and refined using phenix.mr_rosetta[51], phenix.autobuild[52], jelly body refinement using REFMAC5[53], phenix.refine, and manual rebuilding using COOT[54]. The TMD–Linker structure was determined by molecular replacement using the TMD model. The final model of the TMD contained residues 49–310 in mol A, residues 47–310 in mol B, 14 monoolein molecules and 16 water molecules. The final model of the TMD–Linker contained residues 43–320 in mol A, residues 44–353 in mol B and 4 monooleinv molecules. Figures were prepared using CueMol (http://www.cuemol.org/ja/) and PyMOL (Schrödinger).

**SEC coupled to multi-angle laser light scattering (SEC–MALLS).** The instrument set-up for the SEC–MALLS experiment consisted of a Shimazu HPLC system connected in series with a Shimadzu SPD-10Avp UV absorbance detector, a Wyatt DAWN HELEOS 8+ light-scattering detector, and a Shodex RI 101 refractive index detector. Analytical SEC was performed at 4 °C on a Superdex 200 increase 10/300 column equilibrated with buffer, containing 20 mM Tris–HCl (pH 8.0), 150 mM NaCl, and 0.01% LMNG. Elution was monitored in-line with the three detectors, which simultaneously measured UV absorption, light scattering, and refractive index. A 658-nm laser was used in the light-scattering measurement. Molecular masses were calculated using the three-detector method[55], as implemented in the ASTRA software package (Wyatt Technology). The UV extinction coefficient of the protein was calculated by PROTPARAM and assumed to be 1.909. The refractive index increments (dn/dc) of the protein and the detergent were assumed to be 0.185 and 0.132, respectively.

**In vitro assay of the enzymatic activity of Rh-PDE.** Phosphodiesterase activity was measured in vitro[13]. HEK293T cells were transfected with the pcDNA3.1-Rh-PDE variant plasmids by the calcium phosphate method. The DMEM/F-12 media contained 0.5 μM all-trans-retinal, penicillin and streptomycin. The cells were harvested after 24 h and washed in buffer A (140 mM NaCl, 3 mM MgCl$_2$, 50 mM HEPES-NaOH, pH 6.5). The cells were resuspended with buffer A and homogenized using a Potter-Elehjem Grinder (Wheaton) and a syringe with a 27 G needle. The syringe was filled and drained five times while stirring the homogenate. Samples were kept in the dark before measurement for at least 2 hours. Catalytic activity was measured at room temperature in 100 μL buffer A with 1.6–1.8 μg (in the case of cGMP) or 16–18 μg (in the case of cAMP) of total HEK293 cell protein, in a 1.5 ml sample tube. The sample was illuminated with a xenon lamp (MAX-303, Asahi Spectra Co., Ltd., Japan) through a Y52 sharp cut filter (Hoya Candeo Optronics, Tokyo, Japan) (7 mW mm$^{-2}$). Reactions were initiated by adding cyclic nucleotides (final concentration 100 μM). Aliquots were taken out at different time points, and the reactions were immediately terminated by adding 100 μL of 0.1 N HCl and the samples were frozen in liquid nitrogen. After thawing, the samples were centrifuged to remove the membranes and denatured proteins. Nucleotides (20 μL aliquots) were separated by HPLC (Shimadzu Systems) with a C18 reversed-phase column (Waters), using 100 mM potassium phosphate (pH 5.9), 4 mM tetrabutylammonium iodide, and 10% (vol/vol) methanol as the eluent. Nucleotides were monitored at 254 nm. Data were evaluated with LabSolutions (Shimadzu). Peak areas were integrated and assigned to the educt cyclic nucleotide, based on the retention time of the corresponding standard compound.

**Immunoblotting of Rh-PDE mutants.** Protein separation was performed on 10% sodium dodecyl sulfate polyacrylamide gel electrophoresis, and immunoblotting (BIO-RAD) was carried out according to standard protocols. The RHO 1D4-tag was detected by an anti-RHO 1D4-tag antibody (The University of British Columbia) diluted to 1:5000 and a HRP-conjugated anti-mouse antibody (GE Healthcare) diluted to 1:10,000 was used as secondary antibody. The chemiluminescent signal was detected by ECL Plus Western Blotting Detection Reagents (GE Healthcare) and quantified with an Image Quant LAS4000 image analyzer (GE Healthcare). Source data are provided as a Source Data file.

**Assay of the enzymatic activity of Rh-PDE in mammalian cells (GloSensor assay).** HEK293 cells were purchased from JCRB Cell Bank and cultured in E-MEM media with ʟ-Glutamine and Phenol Red (Wako), containing 10% (vol/vol)

fetal bovine serum (FBS) and penicillin–streptomycin. The cells were co-transfected with the plasmid carrying the Rh-PDE variant and the pGloSensor-22F cAMP vector (Promega), by using Lipofectamine 2000 (Invitrogen). Changes in the intracellular cAMP concentration of the HEK293 cells were measured by the GloSensor assay (Promega). The transfected cells were incubated with or without 0.5 μM all-trans-retinal (Toronto Research Chemicals). Before measurements, the culture medium was replaced with a $CO_2$-independent medium, containing 10% (vol/vol) FBS and 2% (vol/vol) GloSensor cAMP stock solution (Promega). The cells were then incubated for 2 h at room temperature in the dark. The intracellular cAMP level was observed by monitoring the luminescence, using a microplate reader (Corona Electric) at 27 °C. The cells were treated with 3.5 μM forskolin (Wako), a direct activator of adenylyl cyclase, to elevate the intracellular cAMP level. The cells were illuminated with a xenon lamp (LAX-103, Asahi Spectra Co., Ltd., Japan) through an interference filter (510 nm). The light intensity was adjusted to 2.28 μW/mm$^2$, and measured by an LP1 power meter (Sanwa Electric Instruments Co., Ltd., Japan).

**Sample preparation of Linker–PDE of Rh-PDE**. The PDE domain of Rh-PDE was subcloned into a modified pEG-CGFP-BC vector, with a C-terminal His$_8$ tag and a TEV protease cleavage site, and expressed in Rosetta2 (DE3) cells. Cells were collected by centrifugation (5000×g, 10 min, 4 °C) and disrupted by sonication in buffer A (50 mM Tris, pH 8.0, 100 mM NaCl, 10 mM MgCl$_2$, 20 mM imidazole). Debris was removed by centrifugation (20,000×g, 30 min, 4 °C) and the supernatant was incubated with Ni-NTA resin (Qiagen) for 30 min at 4 °C. The resin was washed with buffer A and eluted with buffer A containing 200 mM imidazole. The eluate was treated with TEV protease and dialyzed against buffer B (50 mM Tris, pH 8.0, 100 mM NaCl, 10 mM MgCl$_2$). The protease and His$_8$ tag were removed with Ni-NTA resin, and the protein was further purified by SEC on a HiLoad 16/600 Superdex 200 pg column (GE Healthcare), equilibrated with buffer B. The peak fractions were collected and concentrated to 10 mg/mL for the following crystallization.

**Crystallization of Linker–PDE of Rh-PDE**. Aliquots of the protein were dispensed onto 96-well sitting drop plates (Swissci) and overlaid with the precipitant solution, using a Gryphon LCP (Art Robbins Instruments, LLC). Crystals were obtained in 29% (v/v) PEG400, 50 mM calcium acetate, pH 5.0, and 200 mM NaCl. All crystals were incubated at 4 °C for 1 week in the dark. Crystals were harvested using micromounts (MiTeGen), supplemented with 15% ethylene glycol as an additional cryoprotectant, and flash-cooled in liquid nitrogen.

**Structure determination of Linker–PDE of Rh-PDE**. X-ray diffraction data sets were collected on beamline BL32XU at SPring-8, with 10 × 15 μm$^2$ (width × height) micro-focused beams and an EIGER X 9 M detector (Dectris). One hundred twenty-eight small wedge data sets (10° per crystal) were collected manually from different potions of a few large crystals (100–300 μm), each of which was broken in cryoloops to have multiple orientations. The collected images were processed in the same way as TMD. In total, 109 datasets were indexed, and finally 64 datasets were selected through hierarchical clustering by using the correlation coefficients of the normalized intensities between datasets[47] and outlier rejections by kamo.multi_-merge. The structure was determined by molecular replacement, using the previously reported PDE domain structure (PDB ID: 5VYD) as the search model with Phenix Phaser[48]. The resultant structure was refined using phenix.refine[56] and manually rebuilt with Coot[54]. The final model of Rh-PDE (Linker–PDE) contained residues 380–708, 1 magnesium ion, 1 zinc ion, 2 PEG molecules, 2 ethylene glycol molecules, and 168 water molecules. Figures were prepared using CueMol (http://www.cuemol.org/ja/).

**High-speed atomic force microscopy**. The full-length Rh-PDE was purified as described above, using buffer C containing 0.03% DDM and 0.006% CHS instead of 0.1% LMNG and 0.01% CHS, and buffer E containing 0.03% DDM and 0.006% CHS instead of 0.01% LMNG and 0.001% CHS.

A laboratory-built HS-AFM operated in the tapping mode was used to image the full-length Rh-PDE. The solubilized proteins were deposited on a freshly cleaved mica substrate and incubated for 5 min. After the incubation, the residual proteins were thoroughly washed with a buffer solution containing detergent (20 mM Tris–HCl, pH 8.0, containing 100 mM NaCl and 0.1% DDM). The HS-AFM observations were performed in the same buffer, at room temperature. The full-length Rh-PDE reconstituted in a lipid bilayer was also observed. The lipid was composed of asolectin from soybean (Sigma-Aldrich, No. 11145), and the details of the reconstitution procedure are described elsewhere[57]. The lipid fractions including the full-length Rh-PDE were deposited on a mica substrate that had been chemically modified with 0.1% (3-aminopropyl)triethoxysilane (Shin-Etsu Silicone, Tokyo, Japan). After 3 min incubation, the sample was washed with the buffer solution without the detergent. The HS-AFM observation was also performed in the non-detergent buffer at a room temperature. All HS-AFM images and movies were processed by plane-fit background subtraction and Gaussian noise-reduction filters.

**Linker modeling and molecular dynamics simulations**. The secondary structure of the Rh-PDE linker region (residues 311–386) was predicted with the PSIPRED 4.0 server[58]. Rosetta 3.11[31] was used for ab initio modeling of the linker region, and 3/9-mer fragments were picked using RamaScore. The best structure from the total of 5000 structures was used for the following MD simulations.

A periodic boundary system, including explicit solvent, was prepared with VMD[59]. The net charge of the simulation system was neutralized through the addition of 150 mM NaCl. The simulation system was 96 × 96 × 96 Å$^3$ and contained 84,698 atoms. The molecular topologies and parameters from the Charmm36 force field[60] were used for the protein and water molecules.

Molecular dynamics simulations were performed with the program NAMD 2.13[61]. The systems were first energy minimized for 1000 steps with fixed positions of the non-hydrogen atoms, and then for another 1000 steps with 10 kcal mol$^{-1}$ restraints for the non-hydrogen atoms. Next, equilibrations were performed for 0.01 ns under NVT conditions, with 10 kcal mol$^{-1}$ restraints for the heavy atoms of the protein. Finally, equilibrations were performed for 0.5 ns under NPT conditions with 1.0 kcal mol$^{-1}$ restraints. In the equilibration and production processes, the pressure and temperature were set to 1.0 atm and 300 K, respectively. Constant temperature was maintained by using Langevin dynamics. Constant pressure was maintained by using the Langevin piston Nosé–Hoover method[62]. Long-range electrostatic interactions were calculated by using the particle mesh Ewald method[63]. The cMD run was performed for 150 ns, to calculate the average total and dihedral potential.

aMD simulations[33,34] were performed for 150 ns, boosting the aMDd, the aMDT, and aMDdual. The dihedral energy threshold $E_{dihed}$ was set at 304 kcal/mol above the average dihedral energy of the cMD, and the dihedral energy boost factor $\alpha_{dihed}$ was set to 60.8 kcal/mol. The total energy threshold $E_{total}$ was set at 13,500 kcal/mol above the average dihedral energy of the cMD, and the dihedral energy boost factor $\alpha_{total}$ was set to 13,500 kcal/mol.

**Reporting summary**. Further information on research design is available in the Nature Research Reporting Summary linked to this article.

## Data availability

The atomic coordinates and structure factors of Rh-PDE have been deposited in the Protein Data Bank (PDB) (https://www.rcsb.org/) with accession codes 7CJ3 (TMD), 7D7Q (TMD–Linker), and 7D7P (Linker–PDE). X-ray diffraction images of TMD-Linker/Linker-PDE and TMD have been deposited in Zenodo (https://doi.org/10.5281/zenodo.4080668) and CXIDB (https://cxidb.org/id-171.html), respectively. Source data of western blotting (Fig. 3c) are provided with this paper. Other data used in this study were available at NCBI with gene ID 16078606 (Rh-PDE gene) and PDB with accession codes 1C3W (bacteriorhodopsin), 1M0M (channelrhodopsin2), 6EID (channelrhodopsin2), 2HD1 (PDE9 catalytic domain), and 5VYD (Rh-PDE PDE domain). All other data are available from the corresponding authors upon reasonable request. Source data are provided with this paper.

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

## Acknowledgements

The diffraction experiments were performed at SPring-8 BL32XU (proposal 2019A2577). We thank the members of the Nureki laboratory and the beamline staff at BL32XU of SPring-8 (Sayo, Japan) for technical assistance during data collection. We also thank Y. Moriwaki for the assistance of the linker modeling. This research was partially supported by the Platform Project for Supporting Drug Discovery and Life Science Research (Basis for Supporting Innovative Drug Discovery and Life Science Research (BINDS)) from AMED under grant number JP19am0101070 (support No. 1627). This work was supported by JSPS KAKENHI grants 16H06294 (O.N.), 17J30010, 30809421 (W.S.), 25104009, 18H03986, 19H04959 (H.K.), 19H05389, 18H04512, 18H01837 (T.U.), 18K06109 (S.P.T.), 19J13316 (T.I.), and by JST PRESTO (JPMJPR1688 to S.P.T.) and CREST (JPMJCR1753 to H.K.).

## Author contributions

T.I. purified and crystallized Rh-PDE, and created an ab initio model of the linker region. W.S. established the purification procedure and initially attempted the crystallization. T.I. and K. Yamashita solved and refined the structure. M.S., K. Yoshida, M.W., K.K., S.P.T., and H.K. performed the spectroscopic analyses. M.S., K. Yoshida, M.W., K.K., and H.K. performed the mutant analyses. T.T. and T.U. performed the AFM analysis. T.I. performed the MD simulations. The paper was mainly prepared by T.I., W.S., H.K., and O.N. W.S., H.K., and O.N. supervised the research.

## Competing interests

The authors declare no competing interests.
