## [Peer Review File · Nature Communications]

REVIEWER COMMENTS

Reviewer #1 (Remarks to the Author):

The manuscript by Ikuta et al. provides three crystal structures of truncated forms of rhodopsin phosphodiesterase (Rh-PDE) which together with mutational studies give insight into the mechanism of this enzyme rhodopsin. The three structures show the transmembrane domain (TMD) as well as the TMD and the PDE domain with connecting linker regions. This allowed modeling of the full length Rh-PDE and gave some insight into light-triggered increase of PDE activity. The work revealed the existence of the predicted additional transmembrane helix (TM0) which is connected to TM1 via an extracellular loop (ECL0). The TMD is more similar to bacteriorhodopsin than channelrhodopsin 2 and shows a dimer arrangement. The crystal structure of TMD with linker region yielded structural insight into half of a the linker region which was modeled by Rosetta to be a three helix bundle. The newly found 8-TM helix topology is important for the function of Rh-PDE because truncation of TM0 and ECL0 affected function and stability.

The work provides very interesting new information on Rh-PDE and is well written and presented. There are some points the authors need to address to clarify and improve the manuscript.

1) Rh-PDE has in the crystal a different dimer interface (TM1 and TM7) compared with channelrhodopsin and heliorhodopsin dimers. The authors write that TMD and TMD+linker structures have a similar interface. It would be good to illustrate in the supplement how similar the interface is in the two structures. Distance mapping might confirm the interface in solution and give valuable information on how much the interface changes upon light activation of Rh-PDE.

2) Mutational analyses were used to investigate the functional role of the N-terminal residues, TM0 and ECL0. The presentation of photoactivity of cAMP and cGMP hydrolysis is not clear. In Figure 3 activities in the dark were normalized whereas no normalization is shown in extended data Figure 4. Normalization is not necessary to see the effect of the truncations on light induced activity change. It would be important to know how much the values reflect differences in expression level (e.g. by western blots) or changes in basal activity. In case of purified Rh-PDE the spectra should give good estimates, also in terms of how much of the protein is present as rhodopsin or opsin, respectively.

3) The authors discussed the functional role of TM0 based on the TMD mutant lacking TM0. The regeneration data in Figure 3b show there is no binding of retinal with both delta-76 and delta-88 constructs in DDM detergent. However, the authors mention that "their enzyme activities retained slight light-dependency, suggesting that the retinal can bind to the truncated mutants" to explain the light dependent activity change in cell. The spectra in DDM indicate the instability of Rh TMD lacking TM0. For interpretation of the light dependency, the authors could have tested the spectral change under the same condition as used for the cAMP/cGMP assay.

Further, the authors should discuss in more detail the cause of 75% loss of retinal binding in DDM and 50% loss of activity for the delta32 mutant which they crystallized. How much effect may the N-terminal deletion have on the wildtype structure?

4) The authors claim a movement of TM7 upon illumination. Is there a possibility to test this claim by other means?

5) Are there ways to test experimentally the folding of the helical bundle of the linker model?

6) The AFM data are interpreted in a way that TMD and PDE domain do not form any direct interaction and their arrangement is flexible. Is it possible to confirm this, e.g. by distance measurements? Is perhaps mica affecting the arrangement of TMD and PDE domain?

7) Distance measurements would provide a higher confidence in the model presented in Fig. 5b. It also would be helpful to show the model rotated by 90 degrees in the supplement to get a better idea how the two domains are oriented towards each other.

8) Is it possible that the light conditions have a large effect on the folding of the linker helix bundle to explain the flexibility between the two domains? Can AFM be measured under dark and light conditions? Is there a difference?

9) Did the authors measure cAMP/cGMP activity of opsin? How does opsin look like in AFM? Such information might be helpful for interpretation of the data.

Minor points:

1) The number of the Lysine residue to which retinal is attached is not consistent. There is a mix of Lys-296 and Lys-297 in text, figures and sequence alignments and needs to be corrected.

2) What parts of the structures were superimposed for calculating RMSD values?

3) Fig. 3b. How has the amount of Rh been normalized?

4) Fig. 4b. Were data obtained in detergent or membranes?

5) Fig. 4c, Number for Glu in figure legend and figure is not the same

6) Fig. 5a-b: What is the difference between a and b? Is one in detergent, the other in membranes?

7) Fig. 5e: Colors are difficult to distinguish

8) Fig. 5i: How much protein was used? How do data compare with wild type?

9) Fig. 7a-c. There is a typo ... model shown in Fig. 4d should read Fig. 4e.

10) In vitro activity assay (page 19): It says that with purified protein 200 ng and 2 g were used for cGMP and cAMP, respectively. There must be a typo. How did the purified samples differ in their rhodopsin/opsin content?

11) Perhaps add in text a short sentence telling what the GloSensor assay is measuring.

12) Methods: Provide reference for expression

13) The authors may discuss the high Rmeas values for the TMD structures.

Reviewer #2 (Remarks to the Author):

The article with ID NCOMMS-20-13793-T is about the structure and function of a rhodopsin phosphodiesterase (Rh-PDE). The authors solved three structures to elucidate the role of the different domains in the protein. Biophysical measurements and mutants contributed to characterise the function of Rh-PDE.

The article is clearly written and well presented. The figures are also clear and well organised.

The research is original and innovative in the field of microbial rhodopsins and it deserves to be published. The novelty is about the characterisation of the structural and functional relationship in this recently discovered (2017) class of enzymes. The authors also clarify that these proteins have an additional helix as predicted, which has a crucial role in the photoactivation of the enzyme. Finally, the authors present a comparison with the other two famous members of microbial opsins (BR and Ch2), highlighting relevant functional differences.

This focused article will work as a starting point for future literature on Rh-PDEs, as well as supporting the understanding of other microbial opsins.

I have a few minor comments.

1) raw 105, 106 and 289, 290.

It is not fully clear to me if a dimer functionally exists. It is clear that the dimer occurs in the crystals. However, as written, the dimer of Rh-PDE seems not so tight (raw 109) and the molecule of monoolein contributes to this.

Later the authors comment that Met 301 and Val 304 can contribute to the dimer, potentially showing a functional characterisation of the dimer. I think that more data are needed to confirm the presence of a functional dimer, as it may be a transient dimer (loosely packed). Moreover, what would be the functional benefits of a dimer for Rh-PDE?

2) I think it is possible to change "becomes dissociated" with "dissociates" in raw 203.

3) The authors mention optogenetics very shortly in the Introduction and Discussion. I think that the argument is valid but it is not expanded. Especially now that the structural and functional information is available, how would this support optogenetics? What can be practically done and why? or which specific question can be addressed now combining your data and optogenetics?

I also think that the research is original by itself and the link with optogenetics is not necessary. In my opinion, the link with optogenetics is slightly overused in papers about photoactivable proteins. Therefore, when used, a more detailed explanation would give more sense.

Reviewer #3 (Remarks to the Author):

This is a wonderful paper. Novel and rich in experiments.

I have following comments for your consideration

1. Prior work of Kumar... Oprian should be discussed in the context of this work (briefly)
2. Refinement for all 3 structure could be improved. Statistics are barely below what would be expected
3. AFM movies could be deleted- they are not informative

Specific comments by Reviewer #1:

1) Rh-PDE has in the crystal a different dimer interface (TM1 and TM7) compared with channelrhodopsin and heliorhodopsin dimers. The authors write that TMD and TMD+linker structures have a similar interface. It would be good to illustrate in the supplement how similar the interface is in the two structures. Distance mapping might confirm the interface in solution and give valuable information on how much the interface changes upon light activation of Rh-PDE.

According to the suggestion, we added Extended Data Figure 5d to compare the interface and Extended Data Figure 5e to show distance mapping of the TMD and TMD-Linker. These figures confirm that TMD and TMD-Linker have a similar interface.

2) Mutational analyses were used to investigate the functional role of the N-terminal residues, TM0 and ECL0. The presentation of photoactivity of cAMP and cGMP hydrolysis is not clear. In Figure 3 activities in the dark were normalized whereas no normalization is shown in extended data Figure 4. Normalization is not necessary to see the effect of the truncations on light induced activity change.

We changed the figure with no normalization (Figure 3e).

It would be important to know how much the values reflect differences in expression level (e.g. by western blots) or changes in basal activity.

We measured the expression level of N-terminal deletion mutants by western blots (Figure 3c, d). Compared to the wild type, the $\Delta 32$, $\Delta 76$ and $\Delta 88$ mutants showed the expression level of 25%, 29% and 14%, respectively.

In case of purified Rh-PDE the spectra should give good estimates, also in terms of how much of the protein is present as rhodopsin or opsin, respectively.

The result will be uninterpretable because purified Rh-PDE loses light-dependency (Figure L1).

Figure L1 Light-dependency of purified Rh-PDE.

3) The authors discussed the functional role of TM0 based on the TMD mutant lacking TM0. The regeneration data in Figure 3b show there is no binding of retinal with both delta-76 and delta-88 constructs in DDM detergent. However, the authors mention that “their enzyme activities retained slight light-dependency, suggesting that the retinal can bind to the truncated mutants” to explain the light dependent activity change in cell. The spectra in DDM indicate the instability of Rh TMD lacking TM0. For interpretation of the light dependency, the authors could have tested the spectral

change under the same condition as used for the cAMP/cGMP assay. Further, the authors should discuss in more detail the cause of 75% loss of retinal binding in DDM and 50% loss of activity for the $\Delta 32$ mutant which they crystallized. How much effect may the N-terminal deletion have on the wildtype structure?

We performed a western blot to measure the expression level of RHO 1D4 tag-fused N-terminal truncated mutants (Figure 3d). The western blot showed that the N-terminal truncated mutants expressed at only 14-29% of Rh-PDE wild-type. Especially, the $\Delta 32$ mutant expressed 25%, which explain that the cause of 75% loss of retinal binding is merely due to the low expression level. Therefore, the reduced activity of the $\Delta 32$ mutant is almost caused by N-terminal deletion. Although the $\Delta 76$ and $\Delta 88$ mutants showed certain expression levels (29% and 14%, respectively), their spectra are quite low at the rhodopsin absorbance (Figure 3b). This result suggests that the $\Delta 76$ and $\Delta 88$ mutants, which lacks TM0, lose the stable retinal binding.

We added the above descriptions into the main text (lines 201-202, 212-213).

4) The authors claim a movement of TM7 upon illumination. Is there a possibility to test this claim by other means?

Time-resolved serial femto-second crystallography (TR-SFX) is a possible method to track the movement upon illumination, but it will cost a couple of years. Therefore we cannot perform TR-SFX for Rh-PDE, but TR-SFX has been adopted for other microbial rhodopsins, such as bacteriorhodopsin (Nango *et al.*, *Science*, 2016), KR2 (Skopintsev *et al.*, *Nature*, 2020) and channelrhodopsin (in review). In the previous many experiments, TM7 was confirmed to move upon illumination in all the rhodopsins, thus it is supposed that TM7 of Rh-PDE also move.

5) Are there ways to test experimentally the folding of the helical bundle of the linker model?

We planned to measure CD spectrum of the linker region exclusively and crystallize it, but our trial failed due to the extremely low expression of isolated linker.

6) The AFM data are interpreted in a way that TMD and PDE domain do not form any direct interaction and their arrangement is flexible. Is it possible to confirm this, e.g. by distance measurements?

According to the suggestion, we have measured the distance between the centers of TMD and PDE domain in the AFM movies and added the plots as Extended Data Figure 8b-d. The measurement confirms that the distance swings around 100 Å and thus the two domains will not form any direct interaction.

Is perhaps mica affecting the arrangement of TMD and PDE domain?

As for the influence of the mica substrate, our previous HS-AFM work demonstrated that oligomeric structures of various microbial rhodopsins reconstituted in a lipid were retained even on the mica substrate (Shibata *et al.*, *Sci. Rep.*, 2018). Also, several soluble proteins were adsorbed on the substrate and imaged by HS-AFM without significant deterioration of both their oligomeric forms and physiological functions (Uchihashi *et al.*, *Nat. Commun.*, 2018 and Mori *et al.*, *Nat. Commun.*, 2018). Thus, we consider that adsorption of Rh-PDE onto the mica substrate does not modulate the higher-order structures. However, since the relative position fluctuations of the TMD and PDE domains seem to be suppressed by adsorption to the substrate, the relative position between the TMD and PDE domains would be more flexible in the solution.

7) Distance measurements would provide a higher confidence in the model presented in Fig. 5b.

As mentioned above, we measured the distance between the centers of the two domains.

It also would be helpful to show the model rotated by 90 degrees in the supplement to get a better idea how the two domains are oriented towards each other.

We appreciate the suggestion and added the rotated model in Figure 5b.

8) Is it possible that the light conditions have a large effect on the folding of the linker helix bundle to explain the flexibility between the two domains? Can AFM be measured under dark and light conditions? Is there a difference?

The light effect cannot be measured because purified Rh-PDE loses its light-dependency, as we mentioned above.

9) Did the authors measure cAMP/cGMP activity of opsin? How does opsin look like in AFM? Such information might be helpful for interpretation of the data.

In addition to the loss of the light dependency mentioned above, opsin with no rhodopsin contamination cannot be prepared because of intrinsic retinal in cells (Yoshida *et al.*, *J. Biol. Chem.*, 2017).

Minor points:

1) The number of the Lysine residue to which retinal is attached is not consistent. There is a mix of Lys-296 and Lys-297 in text, figures and sequence alignments and needs to be corrected.

We apologize for the confusion and corrected to Lys296 (line 133).

2) What parts of the structures were superimposed for calculating RMSD values?

We added a note at every RMSD values in the main text to clarify that C α atoms were used.

3) Fig. 3b. How has the amount of Rh been normalized?

The figure shows non-normalized spectrum.

4) Fig. 4b. Were data obtained in detergent or membranes?

The PDE domain does not have any membrane-embedded part, thus the data is obtained neither in detergent nor membrane.

5) Fig. 4c, Number for Glu in figure legend and figure is not the same

We apologize for the confusion and corrected to Glu478 (line 409).

6) Fig. 5a-b: What is the difference between a and b? Is one in detergent, the other in membranes?

Both snapshots are taken in lipid bilayer. To avoid confusion, we have removed panel mark.

7) Fig. 5e: Colors are difficult to distinguish

We apologize for the coloring and changed it.

8) Fig. 5i: How much protein was used? How do data compare with wild type?

We changed the figure with unnormalized one. Now the figure can be directly compared with Figure 3e.

9) Fig. 7a-c. There is a typo ... model shown in Fig. 4d should read Fig. 4e.

We appreciate the indication and fixed the typo (line 683).

10) In vitro activity assay (page 19): It says that with purified protein 200 ng and 2 g were used for cGMP and cAMP, respectively. There must be a typo. How did the purified samples differ in their rhodopsin/opsin content?

We apologize for the typo. The letter 'μ' was missing for the latter. However, the sentence should be deleted because there is no measurement using purified protein in the main text. Finally, we have deleted the sentence.

11) Perhaps add in text a short sentence telling what the GloSensor assay is measuring.

We added a short description of GloSensor assay (lines 301-302).

12) Methods: Provide reference for expression

We added a reference (Goehring *et al.*, *Nat. Protoc.*, 2014) at line 428.

13) The authors may discuss the high Rmeas values for the TMD structures.

Rmeas values reflect a quality of unmerged data, not merged data that are used for structure refinement. We merged many crystals, and the merged data quality measured by CC1/2 was sufficiently high (Karplus and Diederichs, *Science*, 2012). This is often seen in microcrystallography (Abe *et al.*, *ACS Nano*, 2017; Shihoya *et al.*, *Nat. Commun.*, 2018; Toyoda *et al.*, *Nat. Chem. Biol.*, 2018; etc).

Specific comments by Reviewer #2:

1) raw 105, 106 and 289, 290.

It is not fully clear to me if a dimer functionally exists. It is clear that the dimer occurs in the crystals. However, as written, the dimer of Rh-PDE seems not so tight (raw 109) and the molecule of monoolein contributes to this.

The PDE domain dimerizes with many hydrogen bonds (Figure L2) as the previous study also reported (Lamarche *et al.*, *Biochemistry*, 2017). Other non-rhodopsin PDEs also form dimers (Huai *et al.*, *Proc. Natl. Acad. Sci. U. S. A.*, 2004), thus PDE domain may need to dimerize to catalyze the reaction. With the rigid dimer formation of the PDE domains, the full-length Rh-PDE will also form dimer under physiological conditions.

Figure L2 Dimer interface of PDE domain.

Later the authors comment that Met 301 and Val 304 can contribute to the dimer, potentially showing a functional characterisation of the dimer. I think that more data are needed to confirm the presence of a functional dimer, as it may be a transient dimer (loosely packed).

We performed SEC-MALLS measurement of the purified Rh-PDE TMD (Extended Data Figure 1d, e). The measurement showed that the former peak corresponds to dimer and the latter corresponds to monomer. We also purified TMD mutants (M301A, V304A and M301/V304A) and the chromatograms shows higher monomer ratio compared to WT (Extended Data Figure 1f).

Moreover, what would it be the functional benefits of a dimer for Rh-PDE?

Rh-PDE forms a dimer in the crystal and supposed to form a dimer under the physiological condition, while Rh-GC catalytic domain is a monomer in the crystal and guessed to be in a monomer/dimer equilibrium under the physiological condition (Kumar *et al.*, *J. Biol. Chem.*, 2017). Rh-GC shows ~5,000-fold light dependency with weak dark activity, which is much higher than that of Rh-PDE (~2-fold). The high monomer ratio in the equilibrium may be associated with the weak dark activity of Rh-GC.

2) I think it is possible to change "becomes dissociated" with "dissociates" in raw 203. We appreciate the correction and changed it (line 216).

3) The authors mention optogenetics very shortly in the Introduction and Discussion. I think that the argument is valid but it is not expanded. Especially now that the structural and functional information is available, how would this support optogenetics? What can be practically done and why? or which specific question can be addressed now combining your data and optogenetics? I also think that the research is original by itself and the link with optogenetics is not necessary. In my opinion, the link with optogenetics is slightly overused in papers about photoactivable proteins. Therefore, when used, a more detailed explanation would give more sense.

Thank you for suggestion. Taking account of the reviewer's comments, we have deleted the following sentence regarding optogenetics. "indicating that there is much room for improvement toward optogenetic applications."

"This study deepens our knowledge about the structures and functions of enzyme rhodopsins, and will facilitate the design of a dark activity-reduced Rh-PDE for optogenetic applications."

Specific comments by Reviewer #3:

1. Prior work of Kumar... Oprian should be discussed in the context of this work (briefly)
We added the below discussion at lines 335-340.

Rh-PDE forms a dimer in the crystal and supposed to form a dimer under the physiological condition, while Rh-GC catalytic domain is a monomer in the crystal and guessed to be in a monomer/dimer equilibrium under the physiological condition (Kumar *et al.*, *J. Biol. Chem.*, 2017). Rh-GC shows ~5,000-fold light dependency with weak dark activity, which is much higher than that of Rh-PDE (~2-fold). The high monomer ratio in the equilibrium may be associated with the weak dark activity of Rh-GC.

2. Refinement for all 3 structure could be improved. Statistic are barely below what would be expected

We suppose relatively high R free value is a concern, especially for TMD and TMD-linker. We tried to improve refinement, but there was no substantial improvement. One possible reason is that the resolution cutoff may be inappropriate, so we performed paired refinement (Karplus and Diederichs, *Science*, 2012). As a result, R free values were improved by inclusion of these high resolution data.

3. AFM movies could be deleted- they are not informative

We added Extended Data Figure 8b-d to track the distance between the centers of the TMD and PDE domain to support our full-length Rh-PDE model (Please also refer our responses to Reviewer #1 comment 6 and 7). Hence, we decided to retain these movies.

REVIEWERS' COMMENTS

Reviewer #1 (Remarks to the Author):

The authors addressed most of my concerns satisfactorily.

Their work provides evidence that TM1/TM7 are involved in forming the dimer interface. DEER distance measurements would have been a possibility to confirm the dimer interface and the proposed TM7 movement upon illumination.

The lack of light-dependency in AFM experiments and PDE activity of purified Rh-PDE weakens the information gained from these experiments. The model in Fig. 5b illustrates the elements of Rh-PDE. The mechanism of signal transmission between Rh and PDE domains, however, remains elusive. But this may be beyond the scope of this manuscript.

Minor point: Extended data Fig. 1f: monomer/dimer labels need to be exchanged.
Fig. 1e: correct typo "theoretical"

Reviewer #2 (Remarks to the Author):

I confirm my overview comment of the first revision. I am positively impressed with this research work and I am fine with the answers provided by the authors. This article has my recommendation for publication.